# Humidity Sensitivity Behavior of CH_3_NH_3_PbI_3_ Perovskite

**DOI:** 10.3390/nano12030523

**Published:** 2022-02-02

**Authors:** Xuefeng Zhao, Yuting Sun, Shuyu Liu, Gaifang Chen, Pengfei Chen, Jinsong Wang, Wenjun Cao, Chunchang Wang

**Affiliations:** Laboratory of Dielectric Functional Materials, School of Materials Science & Engineering, Anhui University, Hefei 230601, China; b51814003@stu.ahu.edu.cn (X.Z.); b51814004@stu.ahu.edu.cn (Y.S.); b51814001@stu.ahu.edu.cn (S.L.); b19201039@stu.ahu.edu.cn (G.C.); b19201035@stu.ahu.edu.cn (P.C.); b20301123@stu.ahu.edu.cn (J.W.); b20201064@stu.ahu.edu.cn (W.C.)

**Keywords:** humidity sensing, impedance-type sensors, organometallic halide perovskite

## Abstract

The CH_3_NH_3_PbI_3_ (MAPbI_3_) powders were ground by PbI_2_ and CH_3_NH_3_I prepared by ice bath method. The humidity sensitive properties of an impedance-type sensor based on MAPbI_3_ materials were systematically studied. Our results indicate that the MAPbI_3_-based sensor has superior sensing behaviors, including high sensitivity of 5808, low hysteresis, approximately 6.76%, as well as good stability. Water-molecule-induced enhancement of the conductive carrier concentration was argued to be responsible for the excellent humidity sensitive properties. Interestingly, the humidity properties can be affected by red light sources. The photogenerated carriers broke the original balance and decreased the impedance of the sensor. This work promotes the development of perovskite materials in the field of humidity sensing.

## 1. Introduction

Perovskites with the formula of AMX_3_ (where A stands for an organic group or inorganic cation with twelve neighboring, X is a halide anion, and M is a metal cation) have been evidenced to be exciting solar absorber materials [1]. Besides the light harvesting performance, these compounds have recently demonstrated several intriguing properties, such as high dielectric constant [2], ferroelecticity [3], photorestriction [4], resistive switching [5], and optical cooling [6]. However, the AMX_3_ perovskites suffer from the major obstacle of chemical and structural instability because they are sensitive to environmental factors. Previous work indicated that the perovskite lattice can interact with the polar water molecules due to the formation of strong hydrogen bonds between water molecules and the halide lattice [7]. This characteristic makes the properties of AMX_3_ heavily dependent on the environment moisture. For example, Alberto García-Fernández et al. [8] investigated the electric properties of CH_3_NH_3_PbI_3_ (MAPbI_3_) in wet and dry environments. The results showed that both capacitance and conductivity in wet condition were several orders of magnitude larger than those in dry condition. This humidity sensitive feature gives the materials tremendous promise as probes for sensing of humidity. Truly, outstanding humidity performances of large sensitivity, remarkable fast response/recovery time, small hysteresis loop, and good linearity were reported in humidity sensors based on Cs_2_PdBr_6_ [9], CH_3_NH_3_PbI_3−*x*_Cl*_x_* [10,11], Cs_2_BiAgBr_6_ [12], and CsPbBr_3_ [7].

MAPbI_3_, being an important member of the AMX_3_ family, has properties that strongly depend on environment humidity [8], indicating that it can be used as a humidity sensing material. Although humidity detection by MAPbI_3_ was attempt by Ilin and co-authors [13], details about its humidity performances have not been studied. Additionally, because of the low-cost solution-based method and cheaper CH_3_NH_2_, perovskite MAPbI_3_ is less expensive than most humidity sensing materials such as CsPbBr_3_ [14], Cs_2_PdBr_6_ [9], and Cs_2_BiAgBr_6_ [12]. Moreover, it was proved that illumination could lead to a giant dielectric constant of lead halide perovskite [2]. Hence, we herein present a systematical investigation on the humidity performances of an impedance-type humidity sensor based on MAPbI_3_.

In this work, MAPbI_3_ was fabricated by grinding CH_3_NH_3_I made from an ice bath with PbI_2_. The humidity sensing performance of the humidity sensor based on perovskite MAPbI_3_ were tested at room temperature in the range of 11–94% relative humidity. Our results show that the MAPbI_3_-based humidity sensor exhibits high humidity sensitivity performance. The possible mechanism of the humidity sensitivity and photoinduced changes were also discussed.

## 2. Materials and Methods

### 2.1. Powder Preparation

Methylamine iodide (MAI) was synthesized from 24 mL methylamine (MA) solution and 10 mL hydroiodic acid (HI) solution [15]. MA solution and HI solution were firstly mixed in a smaller beaker that was placed in a larger beaker at 0 °C. The mixed solution was continuously stirred in an ice bath for 120 min until evenly mixed to form a slightly yellowish solution. Then, the resulting mixture was evaporated in a vacuum oven at 90 °C for 7 h until a white precipitate was obtained. After naturally cooling down to room temperature, the obtained precipitate was washed 3 to 4 times with ether in a centrifuge. Eventually, the precipitate was dried at 60 °C for 24 h to obtain white crystalline powder MAI.

Preparation of lead methyl iodide triiodide (MAPbI_3_). An equal molar amount of MAI powder and PbI_2_ powder were ground in an agate mortar for 20 min until a visually black powder was obtained.

### 2.2. Characterization of Materials

The crystalline phase composition and structure of all powders were examined by X-ray diffraction (XRD, Rigaku Smartlab Beijing Co. Beijing, China) using Cu K α radiation. The morphology of MAPbI_3_ powders was analyzed by scanning electron microscope (SEM, Model S 4800, Hitachi Co, Tokyo, Japan). The possible ferroelectricity of MAPbI_3_ was investigated by measuring polarization-electric field (*P-E*) hysteresis loop using a TF analyzer (MultiFerroicII, Radiant technologies Inc, Albuquerque, New Mexico). In doing so, Au top electrodes were sputtered onto the surface of MAPbI3 film to form a sandwich structure of Au/MAPbI_3_/Au. The ultraviolet photoelectron spectroscopy (UPS) measurement was carried out using He-I radiation (21.2 eV) in vacuum (10^−8^ mbar) to measure the highest filled energy level of valence band (VBM) of MAPbI_3_ film. The sample was biased at −10 V for measurement in the secondary electron cut off region.

### 2.3. Humidity Sensor Fabrication and Performances Measurements

The MAPbI_3_-based humidity sensor was fabricated by aerosol deposition method on Al_2_O_3_ substrate covered with Au interdigitated electrodes. Figure 1 presents the photo images of the Al_2_O_3_ substrate and fabricated sensor. The sensor fabrication processes are as follows: firstly, a little amount of MAPbI_3_ powder was mixed with anhydrous ethanol, and ultrasound for 15 min to form a homogeneous paste. Then, the paste was uniformly sprayed onto a Al_2_O_3_ substrate using a 0.2 mm caliber spray pen (Sao Tome V130). Two copper wires were fixed on the electrodes with silver glue. Finally, the sensor was dried at 100 °C for 10 min and naturally cooled down to indoor temperature. The whole process of sample preparation was shown in Figure 2.

The different relative humidity (RH) environments were obtained by saturated salt solutions of LiCl, MgCl_2_, Mg (NO_3_)_2_, NaCl, KCl, and KNO_3_ in closed containers. The environments above these solutions can provided RH levels of 11, 33, 54, 75, 85, and 94%, respectively. The impedance of the sensor was measured by an impedance analyzer (Hioki 3532-50 LCR). The experimental setup was illustrated in Figure 3. The influence of illumination on the MAPbI_3_-based sensor was reflected by impedance variation with and without light provided by 10 small red LED lights.

## 3. Results

### 3.1. Structure and Morphology Characterizations

The XRD pattern of the MAPbI_3_ powder was shown in the Figure 4a. The peaks at 14.1°, 28.4°, 31.7°, 40.5°, and 43.0° are assigned to (110), (220), (310), (224) and (314) planes, respectively. These peaks agree perfectly with MAPbI_3_ perovskite tetragonal structure (The COD ID of MAPbI_3_ is 2107954) [16]. The average grain size of the MAPbI_3_ powder was clarified to be ~1.39 μm from the SEM image shown in the inset of Figure 4a. Figure 4b shows the *P-E* loops of the MAPbI_3_ measured at room temperature with a frequency of 10 Hz under various voltages of 15, 30, and 45 V. The thickness of the film is 450 mm. The ellipse-shaped *P-E* loops indicate that the MAPbI_3_ exhibits no ferroelectric behavior macroscopically at room temperature. This result is consistent with that reported in Refs. [3,17].

### 3.2. Humidity Sensitive Properties

The humidity sensitive performance of the MAPbI_3_-based resistive sensor was studied by testing the impedance upon exposing the sensor to various RH levels with the a duration time of 5 min in each level. Figure 5a presents the impedance as a function of RH level of the MAPbI_3_-based sensor at different frequencies. The impedance decreases with increasing measurement frequency. The curve measured with 100 Hz shows the largest impedance variation. Hence, 100 Hz is chosen as the optimum measuring frequency and will be used in the following part.

Based on the data in Figure 5a, the humidity sensitive response (S) of the sensor can be calculated according to the relation [18,19]:S = *Z*_d_/*Z*_h_(1)
where *Z*_d_ and *Z*_h_ are the impedance values measured at 11%RH and at a specific RH level, respectively. The MAPbI_3_-based sensor showed a superior sensitivity of S = 5808. Figure 5b displays a contrast of sensitive response between the MAPbI_3_-based sensor and other impedance-type sensors reported in the literature [12,20,21,22,23,24]. The comparison highlights that the MAPbI_3_ shows the largest sensitivity.

For the purpose of fully characterizing the performance of the MAPbI_3_-based sensor, the hysteresis and recovery/response curves of the MAPbI_3_-based sensor were tested at 100 Hz. The results were given in Figure 6a,b, respectively. The hysteresis curve was acquired by switching the sensor between the containers with the different RH levels of 11, 33, 54, 75, 85 and 94% in turn, and then shifting back. After an exposure duration of 5 min in each of the RH levels, the impedance was recorded under the optimum frequency of 100 Hz. Based on the measured impedance values, the humidity hysteresis values can be reckoned by the following formula [25]:(2)$$\left[\frac{\log(Z_{\mathrm{ads}})-\log(Z_{\mathrm{des}})}{\log(Z_{\mathrm{ads}})}\right]\times 100\%$$
where *Z*_des_ and *Z*_ads_ represent the impedance value of the desorption and the adsorption processes, respectively. A hysteresis value of 6.76% is obtained at 11%RH for the MAPbI_3_-based sensor. The physical adsorption, which is toilless to be desorbed in MAPbI_3_ material, is much larger than the chemical adsorption that can be weakly influenced by environment humidity at low humidity level, which results in good hysteresis behavior of the sensor at low RH levels. Studies have shown that there is a steep increase in the water uptake in the RH level beyond 80%RH [26]. Therefore, the sensor still exhibits notable hysteresis under high humidity levels because the physical adsorption is greatly increased. Figure 6b displays the response and recovery time for the MABbI_3_-based sensor recorded between 11% and 94%RH. The result shows that the response and recovery times are 31 s and 148 s, respectively. Response/recovery time under other relative humidity levels are listed in Table 1.

The repeatability of the sensor was measured by switching the sensor between 11 and 94%RH levels for 5 cycles. The stability test was conducted over a period of 135 days. The results of repeatability and stability were shown in Figure 7a,b, respectively. After 5 cycles, the impedance under 11%RH remains 88.13% of the initial value, while under 11%RH, the impedance hardly changes. This result indicates that the sensor shows satisfactory repeatability. The impedance curves shown in Figure 7b remain almost constant, revealing that the sensor exhibits good long-term stability.

### 3.3. Influence of Light on the Humidity Sensing Properties

Previous work has indicated that red light affects the electric properties of MAPbI_3_ [26]. To investigate the influence of illumination on the humidity sensing properties of the MAPbI_3_-based sensor, 10 red LED lights were used as light source. The humidity sensor was placed in the environments of 11%, 54%, and 94%RH successively, and the impedance values were recorded during the three stages of darkness–lightness–darkness with the duration time of 300 s in each stage. The results were plotted in Figure 8. It is clearly shown that under a low humidity level of 11%RH, the impedance can be notably reduced by the light because the photogenerated carrier breaks the original carrier balance and increases the electron concentration, leading to the enhancement of electron conductivity. In the medium and high humidity levels, a similar situation of illumination-induced impedance decrease to that of the low humidity level was observed when the lights were on. However, this impedance variation becomes much weaker than that in 11%RH. However, in the later stage of lighting, the electron concentration decreases due to electron–hole combination, so the impedance increases again. After the light source was switched off, considering that the non-equilibrium carriers does not disappear immediately, the impedance value increases at low and medium humidity levels due to the electron–hole interaction. However, impedance remarkably decreases at high humidity level, probably because the interaction between the light source and the continuous layer of water that forms on the surface of the sample.

## 4. Discussion

To investigate the sensing mechanism, complex impedance diagrams of MAPbI_3_-based sensor were measured with different RH levels. Figure 9 displays the complex impedance plots (Nyquist plots) obtained in different RH environments. At the low RH levels of 11 and 33%, the Nyquist plots are arc-like curves with large radius. The large radius indicates the high resistance at low humidity levels. The arc shrinks with the increasing of RH from 33% to 54%. A whole semicircle followed by a liner line in the low-frequency range is visible within the measuring frequency range when the RH beyond 75%RH. Further increasing the RH level, the semicircle becomes smaller, but the linear line becomes longer. This phenomenon indicates that both the resistance and reactance underwent a large decrease as the RH increases.

Previous work has manifested that the absorbed water molecules can deform the crystal lattice. As the content of water molecules increases to one molecule in one unit cell, this deformation becomes prominent, but the crystal structure is still not broken down [11]. This implies that perovskite materials have good capability accommodating water molecules. Figure 10a presents the UPS spectrum of the material, where the VBM is 1.35 eV below Ef, and its secondary electron truncation is located at 19.5 eV. In addition to the band gap value calculated from the absorption spectrum, the band structure is acquired and depicted in Figure 10b (the Eg of MAPbI_3_ is 1.55 eV), where a standard *n* type feature could be concluded. With more water molecules absorbed by the structure, more electrons would be injected. This would lift the E_f_ of the material and enhance the conductivity of the material [11]. However, perovskite MAPbI_3_ would form the hydrate-CH_3_NH_3_PbI_3_·H_2_O under high humidity [27]. It is clearly shown in Figure 5a that the impedance of MAPbI_3_-based sensor decreases rapidly at low and moderate humidity levels, while the impedance value decreases slowly at high humidity level. Besides, the color of MAPbI_3_-based sensor changes from black to yellow and then back to black when the sensor was successively exposed to 11, 94%RH and then back to 11%RH for 5 min at each RH level, which indicates the degradation is reversible. Therefore, under high humidity, most of the water molecules entering the crystal lattice were used to form hydrate, which leads to slow increase of electrons in the conduction band and slow increase of electrical conductivity. Moreover, due to the limitation of perovskite crystal structure, the hydrate formed is limited, so the electrical conductivity of this material increases slowly.

## 5. Conclusions

In this work, we present investigation on the humidity properties on the MAPbI_3_-based sensor. The results show that the sensor exhibits high sensitivity, superior to that of many perovskite materials. The humidity performances were argued to be caused by the fact that water molecules entered the perovskite lattice, serving as a strong *n*-type dopant that greatly enhances the concentration of conductive electrons. Our results emphasize that the MAPbI_3_ perovskite has a great promise as humidity sensing material and could have widespread applications in humidity sensor.

## Figures and Tables

**Figure 1 nanomaterials-12-00523-f001:**
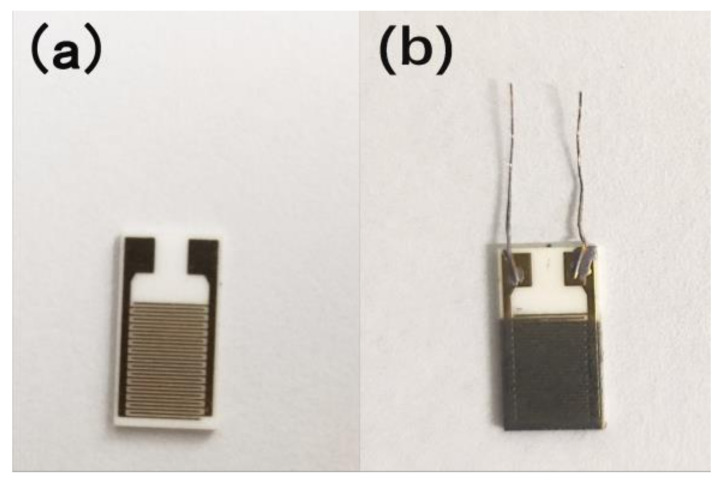
(**a**) Photo images of the Al_2_O_3_ substrate with Au interdigitated electrodes; and (**b**) the MAPbI_3_-based humidity sensor.

**Figure 2 nanomaterials-12-00523-f002:**
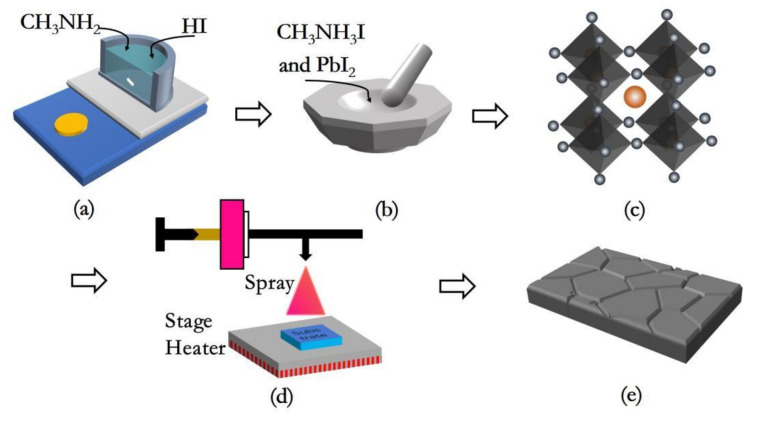
(**a**) Ice bath method; (**b**) grinding; (**c**) the structure of MAPbI_3_ powder; (**d**) aerosol deposition method; and (**e**) the polycrystalline layer structure of MAPbI_3_ thin film.

**Figure 3 nanomaterials-12-00523-f003:**
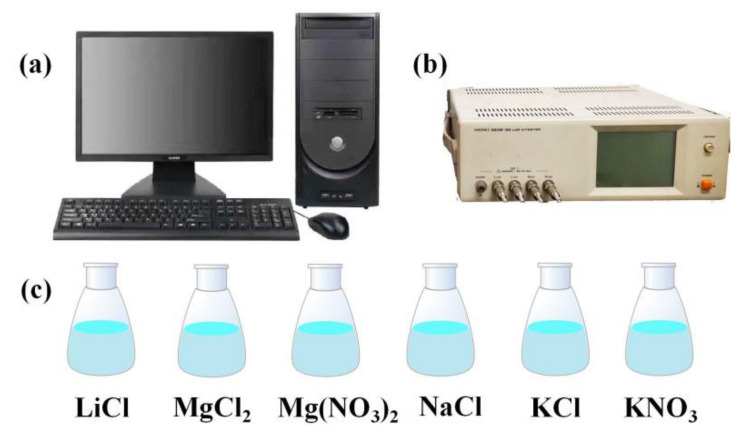
Schematic of the humidity sensing experimental setup: (**a**) PC; (**b**) Hioki 3532-50 LCR; and (**c**) different humidity environments.

**Figure 4 nanomaterials-12-00523-f004:**
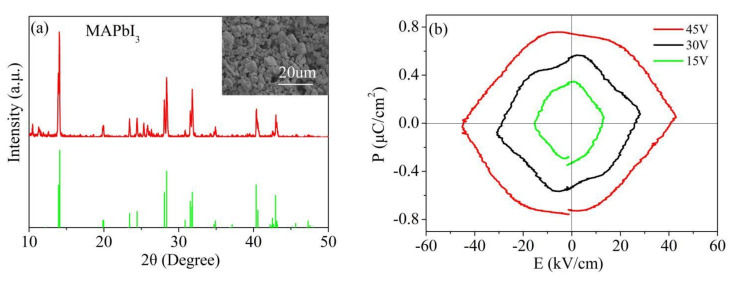
(**a**) XRD pattern and SEM image of MAPbI_3_ powder; and (**b**) *P-E* loops of MAPbI_3_ film measured at the frequency of 10 Hz under the voltage of 15, 30, and 45 V.

**Figure 5 nanomaterials-12-00523-f005:**
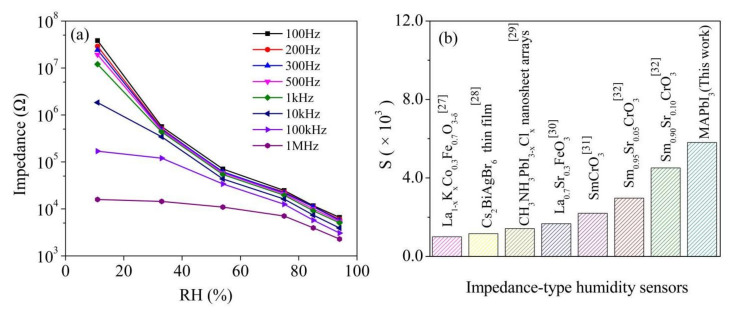
(**a**) The impedance as a function of RH level of the MAPbI_3_-based sensor at different frequencies; (**b**) the contrast between the MAPbI_3_-based sensor and other impedance-type humidity sensors in published literature.

**Figure 6 nanomaterials-12-00523-f006:**
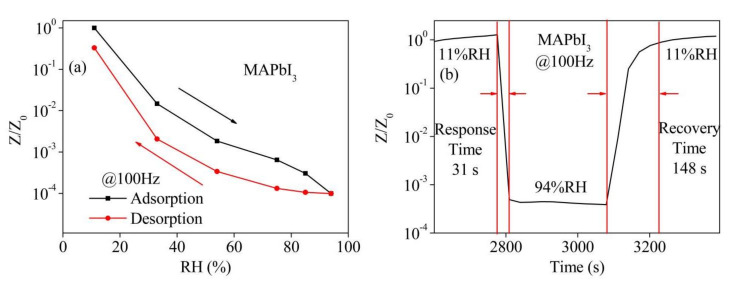
(**a**) Hysteresis behavior and (**b**) recovery/response time of the MAPbI_3_-based sensor (*Z*/*Z*_0_ is the normalized impedance to the initial impedance *Z*_0_).

**Figure 7 nanomaterials-12-00523-f007:**
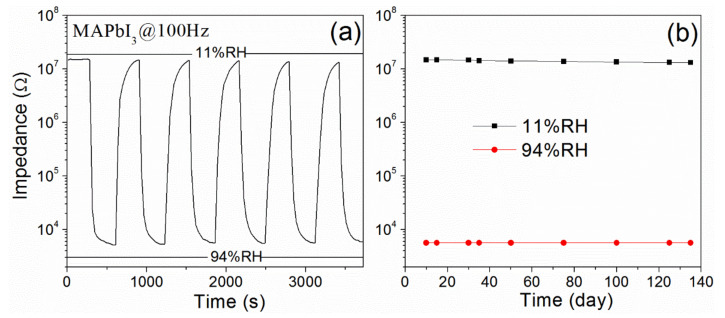
Repeatability (**a**) and long-term stability (**b**) of the MAPbI_3_-based sensor.

**Figure 8 nanomaterials-12-00523-f008:**
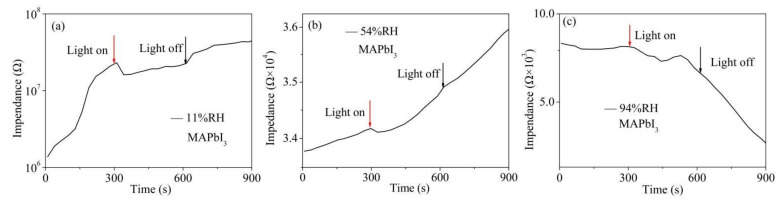
Illumination-induced impedance variations of the MAPbI_3_-based sensor in the environment of (**a**) 11%, (**b**) 54%, and (**c**) 94%RH.

**Figure 9 nanomaterials-12-00523-f009:**
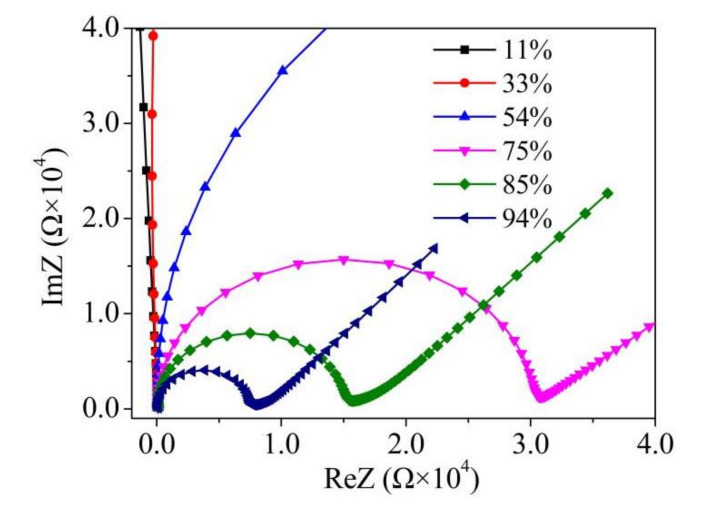
Complex impedance diagrams of the MAPbI_3_-based sensor with different RH levels.

**Figure 10 nanomaterials-12-00523-f010:**
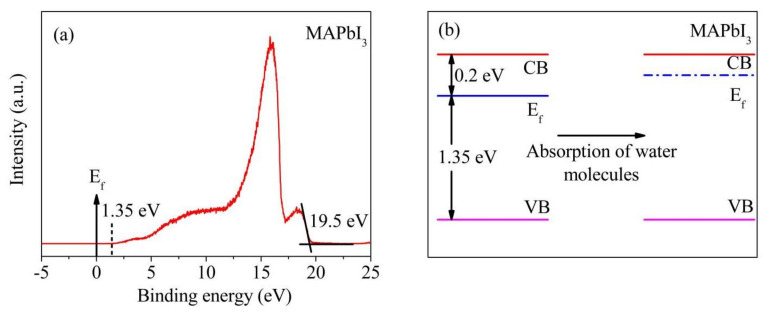
(**a**) The direct UPS spectrum of the as-grown bare MAPbI_3_; and (**b**) the energy level diagrams of as-grown MAPbI_3_ in the cases before and after absorbing water molecules.

**Table 1 nanomaterials-12-00523-t001:** Hysteresis and response/recovery time under different RH levels of the MAPbI_3_-based sensor.

	Hysteresis	Response Time (s)	Recovery Time (s)
MAPbI_3_-based sensor	Max = 18.61% (75%RH) Min = 6.76% (11%RH)	64 (11%→33%)	90 (33%→11%)
		56 (11%→54%)	89 (54%→11%)
		77 (11%→75%)	131 (75%→11%)
		56 (11%→85%)	134 (85%→11%)
		31 (11%→94%)	148 (94%→11%)

## Data Availability

The data is available on reasonable request from the corresponding author.

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
