# Peer review of "Humidity Sensitivity Behavior of CH3NH3PbI3 Perovskite"

_nanomaterials, 2022, doi:10.3390/nano12030523_

Round 1
Reviewer 1 Report
This paper presents an impedance-based humidity sensor based on CH3NH3PbI3 Perovskite. This journal is about nanomaterials but the material part is weak in this study. I do believe that this paper is more suitable to Sensors journal of MDPI. Belows are my comments.
- The material for humidity sensing of this work has a very good performance. Was the protocol for preparing this material adopted from the previous paper or proposed by authors?
- The surface of the sensor should be observed by SEM to evaluate the thickness and the distribution of MAPbI3
- The is no figure about experimental devices and related descriptions.
- In literature, the reviewer found a 2021 paper “High Sensitivity Humidity Detection Based on Functional GO/MWCNTs Hybrid Nano-Materials Coated Titled Fiber Bragg Grating”. Please review this paper and make a comparision with the proposed sensor
- 1 just show images of the sensor, but this journal is about nanomaterials. The authors should provide a comprehensive schematic images
- 5 shows the repeatability of the sensor, however at least five cycles should be shown.
- The long-term stability of the materials should be considered.
- The cost of this material should be considered with others.
- I suggest the authors to carefully revise the paper and emphasize the material part with proper tests and explanations.
Reviewer 2 Report
The manuscript of ‘Humidity Sensitivity Behavior of CH3NH3PbI3 Perovskite’ authored by Zhao et al. demonstrates the humidity properties on the MAPbI3-based sensor. The authors claimed that the sensor shows high sensitivity, which is caused by the finding that water molecules enter the perovskite lattice functioning as a strong n-type dopant, thus, enhance the concentration of conductive electrons. This work is interesting and well presented. Overall, the experimental data can support the hypothesis. I recommend publishing it in Nanomaterials after the authors consider below minor comments.
- There are some typos and errors in the manuscript, e.g., ‘Fig X’ should be ‘Fig. X’, and in Line 66 no need to add ‘and’. In Fig. 8a, it should be ‘Binding energy’ and the unit of Intensity should be ‘a.u.’.
- The authors claimed the different humidity level could influence the perovskite lattice but lack of evidence. Humidity dependent XRD measurements could be considered to probe the water molecules influenced perovskite structure.
- Line 214, ‘This would lift the Ef of the material and enhance the conductivity of the material.’ Can the authors provide any reference for this claim?
Round 2
Reviewer 1 Report
I have no further comments
Author Response
Yes, we have done as per your suggestion. Please see the changes which are marked red in the revised text.